# Crop Residue Return Rather Than Organic Manure Increases Soil Aggregate Stability under Corn–Soybean Rotation in Surface Mollisols

**Yang Xiao** [1,2], **Meng Zhou** [1,2,*], **Yansheng Li** [2], **Xingyi Zhang** [2], **Guanghua Wang** [2], **Jian Jin** [2], **Guangwei Ding** [3], **Xiannan Zeng** [4] **and Xiaobing Liu** [1,2,*]

1   College of Resources and Environment, Northeast Agricultural University, Harbin 150030, China; xiaoxiaoyang22222@163.com
2   Key Laboratory of Mollisols Agroecology, Northeast Institute of Geography and Agroecology, Chinese Academy of Sciences, Harbin 150081, China; liyansheng@iga.ac.cn (Y.L.); zhangxy@iga.ac.cn (X.Z.); wanggh@iga.ac.cn (G.W.); jinjian@iga.ac.cn (J.J.)
3   Chemistry Department, Northern State University, Aberdeen, SD 57401, USA; Guangwei.ding@northern.edu
4   Institute of Crop Cultivation and Tillage, Heilongjiang Academy of Agricultural Sciences, Harbin 150023, China; zengxiannanzxn@163.com
*   Correspondence: zhoumeng@iga.ac.cn (M.Z.); liuxb@iga.ac.cn (X.L.)

**Abstract:** Fertilization practices change soil organic carbon content and distribution, which is relevant to crop rotation and soil aggregates. However, how fertilization management under corn–soybean rotation affects soil organic carbon and aggregate stability at different soil depths in Mollisols is unclear. The effects of 6–yr fertilization under corn–soybean rotation on aggregate stability, soil organic carbon content and storage, and size distribution in soil aggregates were investigated. Five different fertilization practices were carried out in 2013: corn and soybean without fertilizer; corn with chemical fertilizer, soybean without fertilizer; corn with chemical fertilizer, soybean without fertilizer, returning the corn and soybean residues; corn and soybean with chemical fertilizer; and corn with chemical fertilizer, soybean with farmyard manure. Compared with corn and soybean without fertilizer, returning the corn and soybean residues increased bulk SOC content, and enhanced mean weight diameter and geometric mean diameter values at 0–10 cm because of increased water–stable aggregates (WSA) larger than 2 mm proportion and decreased $WSA_{<0.053mm}$ proportion. Simultaneously, corn with chemical fertilizer and soybean with farmyard manure increased bulk soil organic carbon content but reduced mean weight diameter and geometric mean diameter values at 0–20 cm due to increased $WSA_{<0.053mm}$ proportion and decreased $WSA_{>2mm}$ proportion. Altogether, the application of consecutive returning crop residues and chemical fertilizer in alternate years is the most favorable approach for soil organic carbon accumulation and aggregate stability at 0–10 cm under corn–soybean rotation in Mollisols.

**Keywords:** crop residue return; manure; aggregate stability; soil organic carbon; Mollisols

## 1. Introduction

Crop growth and development are heavily dependent on soil structure. Soil aggregates and their stability are regarded as indicators of soil structure, which is identified by mean weight diameter (MWD), geometric mean diameter (GMD), and fractal dimension (D) [1]. Aggregates are classified by particle size in the soil [2]. Different sizes of aggregates, especially macroaggregates (>0.25 mm) protect soil organic carbon (SOC) from decomposing and degrading [3]. As the binding agent and the kernel during the formation of soil aggregates, SOC content is usually related to the number of aggregates [2,4]. In addition, SOC is an indicator of soil fertility, which is an essential part of agricultural production in bonding with soil physicochemical properties, for instance soil aeration, water content, and tillage [5]. Filho et al. confirmed that soil is less aggregated in the lower organic

carbon regions [6]. The amount of crop residue returned to the soil and the chemical substances released from plants also influence the stability, rate of formation, and turnover of aggregates [7,8].

Crop rotation is used to attenuate the drawbacks of agricultural intensification, as it can increase soil carbon accumulation, enzyme activity, and microbial biomass, especially when cover crops are included [9,10]. Corn–soybean rotation has been found to promote corn yield and economic profit [11]. Increased soil available N and microbial diversity, but reduced N fertilizer requirements and crop disease incidence with corn–soybean rotation compared to continuous corn or soybean have also been found [12–14]. However, negative impacts have been reported, such as declines in soil C due to the reduced efficiency by plant residues retained as soil C [15], and reduced soil aggregate stability [16]. Fertilization practices also have impacts on soil aggregates. In the past decades, organic farming has been recommended based on the merits of environment–friendly and agriculture–sustainable [17]. Studies have shown that soils with the application of organic manures illustrate a smaller bulk density, but larger water holding capacity, and greater organic matter content [18,19]. In cultivated soils, adding manure often demonstrates a better soil physical condition and a higher MWD value [20–22]. As a measure of increasing organic matter input, crop residue return can relieve resource waste and increase inputs of organic C and other nutrients, including nitrogen (N), phosphorus (P), and potassium (K) [23–26]. On average, straw returning increased SOC sequestration by 12.8% ± 0.4% and enhanced soil macroaggregates [27].

As one of the four continuous regions in Mollisols throughout the world, Chinese Mollisols are distributed in Northeast China, which accounts for 11.9% of the world's total, with an area of $109 \times 10^4$ km$^2$ [28]. The Mollisol region in Northeast China is the country's "bread basket" and ecological parclose due to its inherently fertile and higher organic matter content ranging from 3–10% [29,30]. However, excessive reclamation and irrational farming management have caused SOC loss and soil erosion problems in the region [31]. Countermeasures such as conservation tillage, crop rotation, and return of organic materials have been developed to deal with the negative effects.

Although soil aggregate stability and SOC in Mollisols have been extensively examined by many researchers [32–34], information linking the two indicators and refining them to different sized aggregates at various soil depths is limited. Our research attempts to comprehend the impacts of different fertilization measures on SOC content/storage in bulk soils and different sized aggregates, and aggregate stability at different depths. We first investigate the SOC changes in Mollisols after 6–yr application of chemical fertilizer, manure, and crop residue return under corn–soybean rotation, and then examine how these changes affect the soil aggregates associated C content/storage, and the changes in soil aggregate stability. The outcome of the research will be of theoretical and practical significance for enhancing soil quality and mechanisms involved between SOC and aggregate stability.

## 2. Materials and Methods

### 2.1. Experimental Site

This study was carried out at the Hailun Station of Soil Erosion Monitoring and Research, Northeast Institute of Geography and Agroecology, Chinese Academy of Sciences (126°49′ E, 47°21′ N) on a typical Mollisol of Northeast China. The site is under semi-humid continental monsoon climate conditions with an effective accumulated temperature of 1030.7 °C, and the average annual rainfall and temperature are 553.9 and 2.4 °C, respectively. The thickness of topsoil is 30 cm, and the basic properties of soil fertility are described in Table 1. Soil pH was determined by a glass electrode pH meter using soil: water ratio of 1:5. The BD was measured using the ring knife method. Soil organic carbon (SOC) and total N (TN) were determined by elemental analyzer. Soil–available phosphorus was measured by the sodium hydrogen carbonate solution–Mo–Sb anti–spectrophotometric method. Soil available potassium was determined by flame spectrophotometers after soil samples were extracted with ammonium acetate.

**Table 1.** Basic properties of soil fertility in experimental area.

| pH | Bulk Density (g cm$^{-3}$) | Organic Matter (g kg$^{-1}$) | Total Nitrogen (g kg$^{-1}$) | Available Phosphorus (mg kg$^{-1}$) | Available Potassium (mg kg$^{-1}$) |
|---|---|---|---|---|---|
| 6.8 | 1.45 | 32.2 | 1.63 | 17 | 248 |

*2.2. Experimental Design and Management*

Our experimental plot was carried out in 2013 with a corn–soybean rotation. Corn was planted in the first year, followed by soybean, and subsequent crops were planted in the same order. The crops were sowed in early May and harvested in early October manually. Each plot includes 12 rows with 9 m row long and 0.7 m inter–row distance. The planting density was 250,000 ha$^{-1}$ for soybean and 48,000 ha$^{-1}$ for corn, according to the local farmers' sowing method.

In total, five different fertilization treatments were carried out. Each treatment included three replications with a completely random block arrangement. The area of each plot was 9 m × 8.4 m.

The five fertilization treatments were:

$C_{NoF}$–$S_{NoF}$: Planting corn without fertilizer, then planting soybean without fertilizer in the next year.

$C_{CF}$–$S_{NoF}$: Planting corn with chemical fertilizer, then planting soybean without fertilizer in the next year.

$C_{CR}$–$S_{NoR}$: Planting corn with chemical fertilizer, returning the corn residues to the field after harvest, then planting soybean without fertilizer in the next year, but returning soybean stalks to the field in middle October.

$C_{CF}$–$S_{CF}$: Planting corn with chemical fertilizer, and planting soybean with chemical fertilizer in the next year.

$C_{CF}$–$S_{FYM}$: Planting corn with chemical fertilizer, and planting soybean with farmyard manure in the next year.

In the $C_{CR}$–$S_{NoR}$ treatment, corn or soybean residues were cut into <3 cm pieces and then ploughed into topsoil mechanically. Residues in other treatments were all removed after harvest. Chemical fertilizer application standards refer to the regular fertilizer application of local farmers. Concretely, N, P, and K (pure N, $P_2O_5$, and $K_2O$) fertilizers were applied to soybean at 50, 30, and 21 kg ha$^{-1}$. Next year, N, P, and K fertilizers were applied to corn at 61.5, 30, and 21 kg ha$^{-1}$ as basal fertilizer. An additional 193.5 kg N ha$^{-1}$ was applied to corn during the jointing period as topdressing fertilizer. The types of fertilizers were urea, diammonium phosphate, and potassium sulfate. The basal fertilizer was applied manually to 20–25 cm soil depth, while the topdressing fertilizer was applied manually to 10 cm soil depth. In the $C_{CF}$–$S_{FYM}$ treatment, naturally decomposed manure in the amount of 15,000 kg ha$^{-1}$ was applied into the 30 cm topsoil by mixing with chemical fertilizer mechanically in middle October when the crops were harvested. The average composition of manure is C with 454 g kg$^{-1}$, N with 20.7 g kg$^{-1}$, P with 7.93 g kg$^{-1}$ and K with 11.8 g kg$^{-1}$ on a dry weight basis [35]. The soybean [*Glycine max* (Merrill.) L.] and corn (*Zea mays* L.) varieties planted were Dongsheng No.1 (mature period 98 days, medium–early maturing variety) and Xingken No.5 (mature period 110–113 days, medium–early maturing variety), respectively. The abamectin and ethazine butyl ester, each with 0.75 kg ha$^{-1}$ were applied manually, and a mechanical tillage operation was implemented.

*2.3. Soil Collecting*

Soil samples were collected in early Oct. 2018 after soybean was harvested, undisturbed soil samples of 0–10 cm, 10–20 cm, and 20–30 cm were sampled by containers sized at 20 cm × 30 cm × 30 cm to determine the size distribution of soil water–stable aggregates (WSA). In each plot, three sample points were diagonally selected, and samples at the same depth were mixed together. A 100 cm$^3$ volume and 5.046 cm diameter ring knife

were applied at the same sampling points and depths for calculating bulk density and SOC content.

### 2.4. Laboratory Method

About 50 g of soil sample was taken and weighed, then dried at $105 \pm 2\,°C$ to constant weight. Soil moisture content was calculated by Equation (1):

$$SMC = (M_1 - M_2)/M_2 \tag{1}$$

*SMC* is soil moisture content; $M_1$ *and* $M_2$ represent wet and dried soil weight, respectively.

Ring knives with soil were dried at $105 \pm 2\,°C$ to a constant weight, denoted as $M_0$ measured by a scale. The bulk density (*BD*) values were calculated via the following Equation (2):

$$BD = M_0 - M_R/V \tag{2}$$

*BD* is the value of bulk density, $M_R$ is the weight of ring knife, and *V* is the volume of ring knife.

Wet sieving was used to determine the distribution of water–stable aggregates (WSA). The size distribution of WSA was measured by the soil aggregates analyzer (TPF–100, Tuopuyunnong, Hangzhou, China). The detailed manipulations were as follows: 50 g soil samples were air–dried for 24 h and evenly distributed over the nested sieve surfaces through a series of three sieves (2 mm, 0.25 mm, 0.053 mm) to isolate four aggregate size fractions. The four aggregate sizes with >2 mm, 0.25–2 mm, 0.053–0.25 mm, and <0.053 mm refer to larger macro–aggregates, smaller macro–aggregates, micro–aggregates, and silt + clay fractions, respectively. The nest was set at the highest point when the oscillation cylinders were filled with distilled water. To slake the air–dried soil, 1 L of distilled water was rapidly added to each cylinder until the soil sample and top screen were covered with water. The soils were submerged in water for 10 min before the wet–sieving action. The oscillation time, stroke length, and frequency were 10 min, 4 cm vertical, and 30 cycle $min^{-1}$, respectively [36]. Finally, the soils remaining on each sieve were collected by Petri dish weighting $M_1$, then drying them at $60$–$80\,°C$ and weighting $M_2$.

The mass of each size aggregate (*M*) was computed by Equation (3). The size distribution of the WSA was based on the M. The proportion of each aggregate size ($P_i$) was computed by Equation (4).

$$M_i = M_2 - M_1 \tag{3}$$

$$P_i = M_i/50 \times 100\% \tag{4}$$

The *MWD*, *GMD*, and *D* were calculated by Equations (5), (6), and (7), respectively.

$$MWD = \sum_{i=1}^{n}(\overline{X_i}\,P_i) \tag{5}$$

$$GMD = \exp\left\{(\sum_{i=1}^{n} P_i \lg\overline{X_i})/(\sum_{i=1}^{n} P_i)\right\} \tag{6}$$

$$(3-D)\lg\left\{\overline{X_i}/X_{max}\right\} = \lg\left\{W_{(\delta \le \overline{X_i})}/W\right\} \tag{7}$$

$\overline{X_i}$ indicates the mean diameter of each size (mm), calculated by the average value of max and min diameter of each sized aggregates; $X_{max}$ is the diameter of soils on the sieve of 2 mm aperture size before wet sieving, which is 10 mm. $W_{(\delta \le \overline{X_i})}$ is the sum of soil weights (size $\le \overline{X_i}$).

Since the inorganic carbon in Mollisols is negligible [37], the total soil carbon measured by the elemental analyzer (FlashEA 1112, Thermofinnigan, San Jose, CA, USA) is approximately equal to the soil organic carbon; in the following text, it appears in the form

of soil organic carbon (SOC). Before measuring SOC by elemental analyzer, soil samples were ground and passed through a 0.25 mm sieve.

The *SOC* storage was calculated by Equation (8).

$$S_i = M_i \times SOC_i \times BD \times H \times 10^{-1} \tag{8}$$

As to this equation, $S_i$ means the *SOC* storage of $i$–size aggregates (Mg ha$^{-1}$), $M_i$ is the mass fraction, while the $SOC_i$ of $i$–size aggregates. The *BD* is the corresponding bulk density (g $\times$ cm$^{-3}$). H is the soil thickness, whose value is 10 cm in this study.

### 2.5. Statistical Analysis

The original values and statistical analysis were carried out by Excel 2019 and SPSS 24, respectively. All graphs were drawn using Sigma Plot 12.5. Tukey's HSD test at the $p < 0.05$ level ($n = 3$) was applied to determine the significant differences among the treatments with one–way variance analysis. The correlations among the measured parameters were determined through linear regression analysis.

## 3. Results

### 3.1. Size Distribution of Water–Stable Aggregates

In general, regardless of fertilization regimes and soil depths, higher proportions of WSA were found in WSA$_{0.25-2mm}$, ranging from 30–40% (Table 2). Among the five fertilization practices, the highest and lowest proportion of WSA sizes did show a different tendency in the three soil depths.

**Table 2.** Size distribution proportion (%) of water–stable aggregates.

| Soil Depths | Aggregate Sizes | Treatments | | | | |
|---|---|---|---|---|---|---|
| | | $C_{NoF}$–$S_{NoF}$ | $C_{CF}$–$S_{NoF}$ | $C_{CR}$–$S_{NoR}$ | $C_{CF}$–$S_{CF}$ | $C_{CF}$–$S_{FYM}$ |
| 0–10 cm | >2 mm | 13.6 $\pm$ 0.21 b | 8.88 $\pm$ 0.15 c | 17.4 $\pm$ 0.22 a | 8.40 $\pm$ 0.12 c | 4.59 $\pm$ 0.11 d |
| | 0.25–2 mm | 41.0 $\pm$ 0.55 c | 45.0 $\pm$ 0.64 a | 42.0 $\pm$ 0.35 bc | 43.5 $\pm$ 0.53 ab | 43.1 $\pm$ 0.66 abc |
| | 0.053–0.25 mm | 7.91 $\pm$ 0.13 c | 12.1 $\pm$ 0.23 b | 12.2 $\pm$ 0.18 b | 16.3 $\pm$ 0.31 a | 13.9 $\pm$ 0.32 b |
| | <0.053 mm | 37.5 $\pm$ 0.28 a | 34.0 $\pm$ 0.42 b | 28.5 $\pm$ 0.36 d | 31.9 $\pm$ 0.52 c | 38.5 $\pm$ 0.35 a |
| 10–20 cm | >2 mm | 5.42 $\pm$ 0.10 c | 6.90 $\pm$ 0.02 c | 6.32 $\pm$ 0.09 c | 28.4 $\pm$ 0.34 a | 9.50 $\pm$ 0.06 b |
| | 0.25–2 mm | 41.5 $\pm$ 0.25 b | 37.6 $\pm$ 0.34 c | 45.2 $\pm$ 0.53 a | 35.9 $\pm$ 0.38 c | 43.3 $\pm$ 0.15 b |
| | 0.053–0.25 mm | 6.61 $\pm$ 0.18 d | 11.6 $\pm$ 0.29 b | 16.7 $\pm$ 0.32 a | 9.20 $\pm$ 0.14 c | 9.09 $\pm$ 0.17 c |
| | <0.053 mm | 46.5 $\pm$ 0.39 a | 43.9 $\pm$ 0.37 b | 31.8 $\pm$ 0.44 d | 26.5 $\pm$ 0.27 e | 38.1 $\pm$ 0.45 c |
| 20–30 cm | >2 mm | 3.36 $\pm$ 0.04 c | 8.20 $\pm$ 0.10 a | 5.70 $\pm$ 0.04 b | 7.60 $\pm$ 0.15 a | 4.20 $\pm$ 0.08 bc |
| | 0.25–2 mm | 39.0 $\pm$ 0.36 d | 49.7 $\pm$ 0.55 a | 32.7 $\pm$ 0.61 e | 46.9 $\pm$ 0.46 b | 43.2 $\pm$ 0.42 c |
| | 0.053–0.25 mm | 16.2 $\pm$ 0.36 bc | 14.3 $\pm$ 0.28 bc | 20.1 $\pm$ 0.26 a | 17.8 $\pm$ 0.34 b | 16.2 $\pm$ 0.27 c |
| | <0.053 mm | 41.4 $\pm$ 0.33 a | 27.9 $\pm$ 0.37 c | 41.6 $\pm$ 0.27 a | 27.8 $\pm$ 0.16 c | 36.3 $\pm$ 0.22 b |

Note: Different lowercase letters following each value represent significant differences for each aggregate size among different treatments at the same soil layer (Tukey's HSD test, $p < 0.05$). $C_{NoF}$–$S_{NoF}$: corn and soybean without fertilizer; $C_{CF}$–$S_{NoF}$: corn with chemical fertilizer, soybean without fertilizer; $C_{CR}$–$S_{NoR}$: corn with chemical fertilizer, soybean without fertilizer, returning the corn and soybean residues; $C_{CF}$–$S_{CF}$: corn and soybean with chemical fertilizer; $C_{CF}$–$S_{FYM}$: corn with chemical fertilizer, soybean with dairy manure.

Specifically, at 0–10 cm soil depth, compared to $C_{CF}$–$S_{CF}$ treatment, the $C_{NoF}$–$S_{NoF}$ and $C_{CR}$–$S_{NoR}$ treatments raised the WSA$_{>2mm}$ proportion by 61.9% and 107% ($p < 0.05$). The $C_{NoF}$–$S_{NoF}$, $C_{CF}$–$S_{NoF}$, and $C_{CF}$–$S_{FYM}$ treatments in comparison with $C_{CF}$–$S_{CF}$ enhanced the WSA$_{<0.053\,mm}$ proportion by 17.6%, 6.58%, and 20.7%, respectively ($p < 0.05$) (Table 2). The $C_{CR}$–$S_{NoR}$ decreased the proportion of WSA$_{<0.053mm}$ by 10.7%, and $C_{CF}$–$S_{FYM}$ decreased the proportion of WSA$_{>2mm}$ by 45.4% compared to $C_{CF}$–$S_{CF}$ ($p < 0.05$). Simultaneously, the $C_{NoF}$–$S_{NoF}$, $C_{CF}$–$S_{NoF}$, and $C_{CR}$–$S_{NoR}$ treatments raised macroaggregates (WSA$_{>0.25mm}$) proportion by 5.20%, 3.82%, and 14.5% compared to the $C_{CF}$–$S_{CF}$ treatment ($p < 0.05$) (Figure 1a).

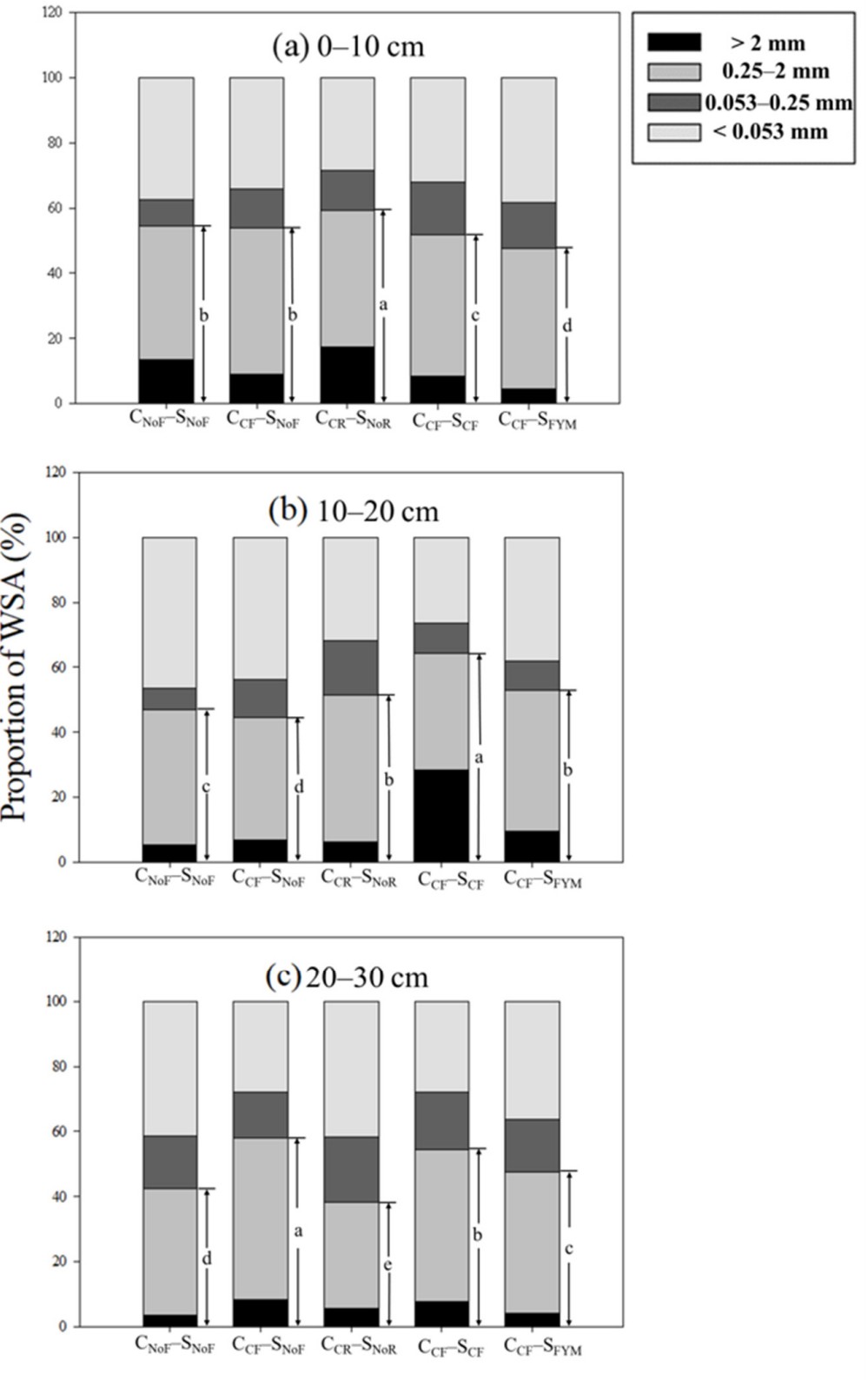

**Figure 1.** Size distribution (%) of water–stable aggregates. Note: Different lowercase letters represent significant differences in the proportions of macroaggregates (>2 mm and 0.25–2 mm) among different treatments at the same soil layer (Tukey's HSD test, *p* < 0.05).

For the 10–20 cm soil depth, compared with $C_{CF}$–$S_{CF}$, the $C_{NoF}$–$S_{NoF}$, $C_{CR}$–$S_{NoR}$, and $C_{CF}$–$S_{FYM}$ treatments enhanced the proportion of $WSA_{0.25–2mm}$ by 15.6%, 25.9%, and 20.6%, and raised the proportion of $WSA_{<0.053mm}$ by 75.5%, 20.0%, and 43.8%, respectively ($p < 0.05$). The $C_{CF}$–$S_{NoF}$ and $C_{CR}$–$S_{NoR}$ treatments increased the proportion of $WSA_{0.053–0.25mm}$ by 26.1% and 81.5% ($p < 0.05$), but the $C_{CR}$–$S_{NoR}$ and $C_{CF}$–$S_{FYM}$ treatments decreased the proportion of $WSA_{>2mm}$ by 77.7% and 66.5% ($p < 0.05$) (Table 2). The $C_{CF}$–$S_{CF}$ treatment exhibited the highest macroaggregates ($WSA_{>0.25\,mm}$) proportion (Figure 1a).

With respect to the 20–30 cm soil depth, the $C_{CR}$–$S_{NoR}$ increased the proportion of $WSA_{0.053–0.25mm}$ and $WSA_{<0.053mm}$ by 12.9% and 49.6% compared with $C_{CF}$–$S_{CF}$ treatment ($p < 0.05$) (Table 2). The macroaggregates ($WSA_{>0.25mm}$) proportion of $C_{CF}$–$S_{NoF}$ was 6.24% higher than $C_{CF}$–$S_{CF}$ ($p < 0.05$) (Figure 1c).

### 3.2. The MWD, GMD, and D Values within Soil Aggregates

The maximal MWD and GMD were presented in the $C_{CR}$–$S_{NoR}$ treatment with 1.46, 0.68 at 0–10 cm depth; and in the $C_{CF}$–$S_{CF}$ treatment with 2.30, 0.81 at 10–20 cm depth, respectively (Figure 2a,b). The lowest MWD and GMD values at 0–10 cm depth were obtained in the $C_{CF}$–$S_{FYM}$ treatment with 0.88, 0.50, and in the $C_{CF}$–$S_{NoF}$ treatment with 0.87, 0.46 at the 10–20 cm depth (Figure 2a,b). With regard to 20–30 cm depth, the GMD values of $C_{CF}$–$S_{NoF}$ and $C_{CF}$–$S_{CF}$ were higher than $C_{NoF}$–$S_{NoF}$, $C_{CR}$–$S_{NoR}$ and $C_{CF}$–$S_{FYM}$ treatments ($p < 0.05$) (Figure 2c). To be specific, compared with the $C_{CF}$–$S_{CF}$ treatment, the MWD and GMD in $C_{CR}$–$S_{NoR}$ treatment were 32.7% and 17.2% higher at 0–10 cm depth ($p < 0.05$) (Figure 2a). As to 10–20 cm soil depth, the MWD and GMD in $C_{CR}$–$S_{NoR}$ treatment were 43.5% and 29.6% lower than $C_{CF}$–$S_{CF}$ treatment ($p < 0.05$) (Figure 2b). Compared with $C_{CF}$–$S_{CF}$, the $C_{CF}$–$S_{FYM}$ treatment reduced MWD value by 20% and 54.8%, and decreased GMD value by 13.8% and 30.9% at 0–10 cm and 10–20 cm depths ($p < 0.05$) (Figure 2a,b). In addition, the $C_{NoF}$–$S_{NoF}$ and $C_{CF}$–$S_{NoF}$ treatments reduced MWD value by 57.0% and 62.2%, and GMD value by 28.4% and 43.2% at 10–20 cm depth in comparison with $C_{CF}$–$S_{CF}$ treatment ($p < 0.05$) (Figure 2b). No significant differences were presented for D value between the two treatments at all soil depths ($p > 0.05$) (Figure 2).

### 3.3. SOC within Bulk Soil in Different Treatments

Among the five fertilization practices, regardless of soil depths, the least SOC content was all obtained in the $C_{NoF}$–$S_{NoF}$ treatment (Table 3). The largest SOC contents were demonstrated in the $C_{CF}$–$S_{FYM}$ treatment with 23.5 g kg$^{-1}$ and 21.8 g kg$^{-1}$ at 0–10 cm and 10–20 cm soil depths.

Compared with $C_{NoF}$–$S_{NoF}$ treatment, the other four treatments increased SOC content at all soil depths ($p < 0.05$), except the $C_{CF}$–$S_{NoF}$ treatment at 0–10 cm soil depth (Table 3). Moreover, compared to the $C_{CF}$–$S_{CF}$ treatment, the $C_{CR}$–$S_{NoR}$ enhanced SOC content by 6.64% and 5.82% at 0–10 cm and 10–20 cm depths, while $C_{CF}$–$S_{FYM}$ enhanced SOC content by 11.4% and 15.3% at 0–10 cm and 10–20 cm depths ($p < 0.05$).

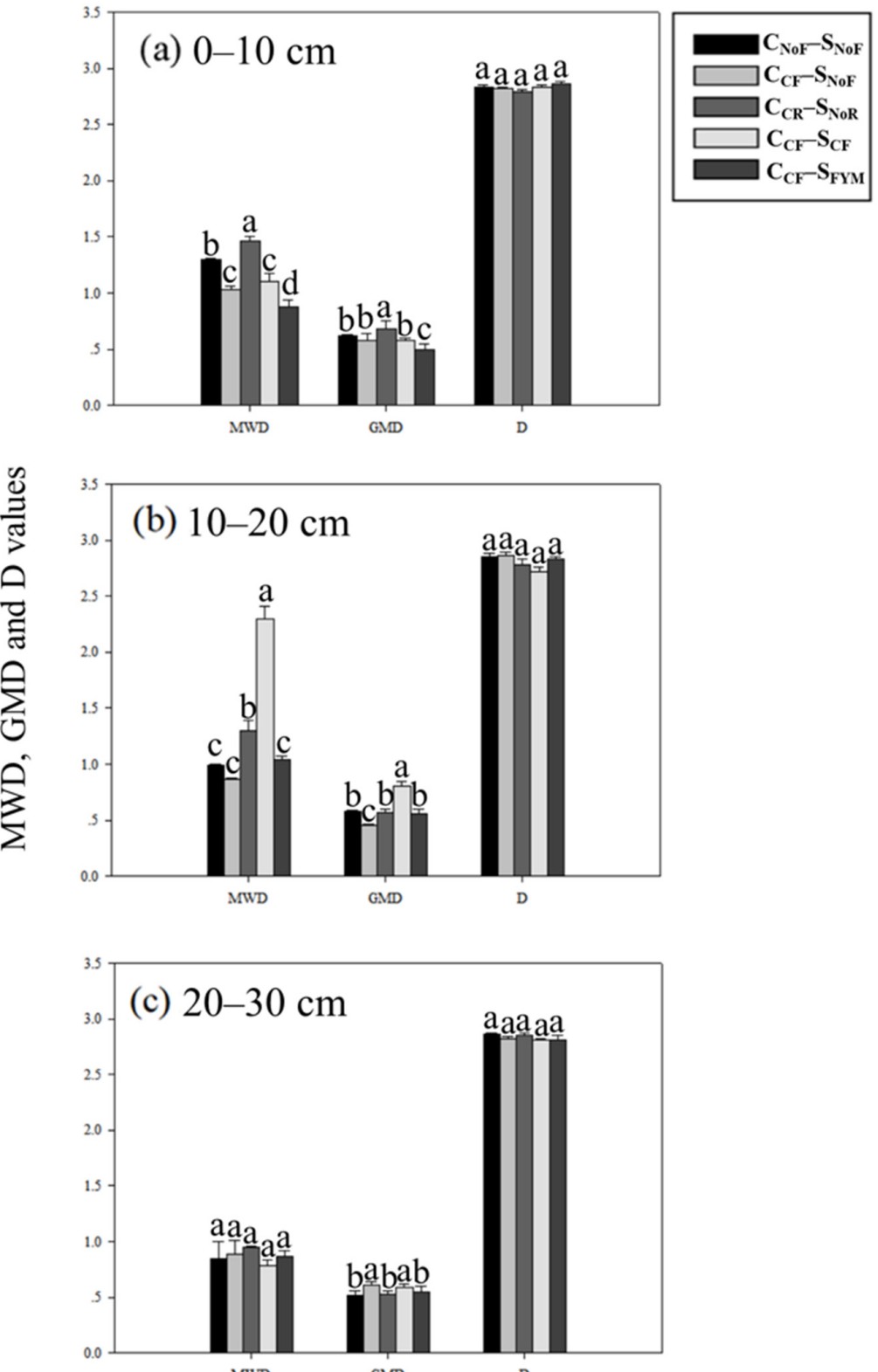

**Figure 2.** MWD, GMD, and D values at different layers. Note: Different lowercase letters represent significant differences among different treatments at the same soil layer (Tukey's HSD test, $p < 0.05$).

**Table 3.** Bulk density, moisture, and SOC contents within bulk soils.

| Soil Depths | Item | Treatments | | | | |
|---|---|---|---|---|---|---|
| | | $C_{NoF}–S_{NoF}$ | $C_{CF}–S_{NoF}$ | $C_{CR}–S_{NoR}$ | $C_{CF}–S_{CF}$ | $C_{CF}–S_{FYM}$ |
| 0–10 cm | Bulk density (g cm$^{-3}$) | 1.30 ± 0.02 a | 1.20 ± 0.01 b | 1.21 ± 0.02 b | 1.27 ± 0.02 a | 1.14 ± 0.01 c |
| | moisture content (%) | 22.5 ± 0.30 d | 22.6 ± 0.16 d | 24.1 ± 0.22 c | 25.6 ± 0.15 b | 26.9 ± 0.23 a |
| | SOC within bulk soils (g kg$^{-1}$) | 19.2 ± 0.28 d | 20.0 ± 0.17 d | 22.5 ± 0.20 b | 21.1 ± 0.22 c | 23.5 ± 0.19 a |
| 10–20 cm | Bulk density (g cm$^{-3}$) | 1.32 ± 0.02 c | 1.38 ± 0.01 b | 1.27 ± 0.01 d | 1.53 ± 0.02 a | 1.23 ± 0.01 e |
| | moisture content (%) | 23.8 ± 0.22 cd | 23.3 ± 0.12 d | 26.5 ± 0.31 a | 24.6 ± 0.14 c | 25.6 ± 0.27 b |
| | SOC within bulk soils (g kg$^{-1}$) | 17.5 ± 0.17 d | 18.6 ± 0.11 c | 20.0 ± 0.21 b | 18.9 ± 0.25 c | 21.8 ± 0.15 a |
| 20–30 cm | Bulk density (g cm$^{-3}$) | 1.52 ± 0.01 bc | 1.55 ± 0.01 b | 1.40 ± 0.01 d | 1.62 ± 0.02 a | 1.50 ± 0.01 c |
| | moisture content (%) | 25.3 ± 0.32 c | 24.8 ± 0.13 cd | 24.0 ± 0.26 d | 27.6 ± 0.17 a | 26.6 ± 0.19 b |
| | SOC within bulk soils (g kg$^{-1}$) | 14.7 ± 0.10 b | 16.0 ± 0.18 a | 15.8 ± 0.14 a | 16.6 ± 0.15 a | 16.1 ± 0.15 a |

Note: Different lowercase letters following each value represent significant differences for each parameter separately among different treatments at the same soil layer (Tukey's HSD test, $p < 0.05$).

### 3.4. SOC Content and Storage within Soil Aggregates in Different Treatments

As a whole, the WSA$_{>2mm}$ exhibited the highest SOC content at each soil depth (Table 4). At 0–10 cm soil depth, compared with $C_{CF}–S_{CF}$ treatment, the $C_{CR}–S_{NoR}$ treatment raised SOC content in WSA$_{0.25–2mm}$ and WSA$_{0.053–0.25mm}$ by 13.4% and 15.6%, while $C_{CF}–S_{FYM}$ treatment raised SOC content in WSA$_{0.053–0.25mm}$ by 23.5% ($p < 0.05$) (Table 4). At 10–20 cm soil depth, the $C_{CF}–S_{FYM}$ treatment raised SOC content in WSA$_{>0.25mm}$ by 8.33% than $C_{CF}–S_{CF}$ ($p < 0.05$). $C_{NoF}–S_{NoF}$ treatment enhanced the SOC content in WSA$_{0.25–2mm}$, WSA$_{0.053–0.25mm}$, and WSA$_{<0.053mm}$ at the 20–30 cm in comparison with the $C_{CF}–S_{CF}$ treatment ($p < 0.05$).

**Table 4.** SOC content (g kg$^{-1}$) within soil aggregates.

| Soil Depths | Aggregate Sizes | Treatments | | | | |
|---|---|---|---|---|---|---|
| | | $C_{NoF}–S_{NoF}$ | $C_{CF}–S_{NoF}$ | $C_{CR}–S_{NoR}$ | $C_{CF}–S_{CF}$ | $C_{CF}–S_{FYM}$ |
| 0–10 cm | >2 mm | 22.2 ± 0.34 a | 19.9 ± 0.40 b | 22.4 ± 0.46 a | 21.4 ± 0.58 ab | 22.9 ± 0.29 a |
| | 0.25–2 mm | 21.3 ± 0.74 abc | 19.4 ± 0.56 c | 22.9 ± 0.35 a | 20.2 ± 0.44 bc | 22.2 ± 0.34 ab |
| | 0.053–0.25 mm | 18.2 ± 0.28 b | 18.4 ± 0.18 b | 20.7 ± 0.29 a | 17.9 ± 0.22 b | 22.1 ± 0.33 a |
| | <0.053 mm | 20.8 ± 0.35 a | 18.0 ± 0.22 b | 20.2 ± 0.34 ab | 17.8 ± 0.41 b | 19.4 ± 0.51 ab |
| 10–20 cm | >2 mm | 19.4 ± 0.06 b | 18.6 ± 0.15 b | 19.3 ± 0.46 b | 19.2 ± 0.34 b | 20.8 ± 0.16 a |
| | 0.25–2 mm | 17.9 ± 0.44 b | 16.8 ± 0.15 c | 18.6 ± 0.35 ab | 19.3 ± 0.17 a | 19.4 ± 0.27 a |
| | 0.053–0.25 mm | 17.0 ± 0.23 ab | 17.0 ± 0.04 ab | 16.4 ± 0.17 b | 17.6 ± 0.06 a | 17.8 ± 0.33 a |
| | <0.053 mm | 16.7 ± 0.33 ab | 17.0 ± 0.42 ab | 16.6 ± 0.21 b | 17.3 ± 0.41 a | 17.3 ± 0.27 a |
| 20–30 cm | >2 mm | 19.5 ± 0.21 a | 14.0 ± 0.33 d | 18.0 ± 0.34 b | 18.9 ± 0.25 ab | 16.1 ± 0.19 c |
| | 0.25–2 mm | 19.8 ± 0.21 a | 13.4 ± 0.10 c | 16.0 ± 0.06 b | 17.0 ± 0.03 b | 14.8 ± 0.20 bc |
| | 0.053–0.25 mm | 19.5 ± 0.04 a | 12.6 ± 0.22 c | 14.8 ± 0.31 bc | 16.0 ± 0.24 b | 13.0 ± 0.10 c |
| | <0.053 mm | 16.9 ± 0.14 a | 10.1 ± 0.19 c | 15.8 ± 0.33 a | 13.0 ± 0.40 b | 10.5 ± 0.06 c |

Note: Different lowercase letters following each value represent significant differences for each aggregate size among different treatments at the same soil layer (Tukey's HSD test, $p < 0.05$).

Generally, the maximum values of SOC storage were all observed in WSA$_{0.25–2mm}$, despite soil depths and treatments (Table 5). Specifically, compared to $C_{CF}–S_{CF}$ treatment,

$C_{CF}$–$S_{FYM}$ increased SOC storage of WSA$_{<0.053mm}$ by 17.9%, 15.4%, and 15.1% at 0–10 cm, 10–20 cm, and 20–30 cm soil depths ($p < 0.05$) (Table 5). However, $C_{CR}$–$S_{NoR}$ increased SOC storage of WSA$_{>2mm}$ by 107% at 0–10 cm depth, WSA$_{0.053–0.25mm}$ by 39.8% at 10–20 cm depth, and WSA$_{<0.053mm}$ by 57.7% at 20–30 cm depth ($p < 0.05$). Besides, at 0–10 cm depth, the $C_{NoF}$–$S_{NoF}$ treatment increased SOC storage of WSA$_{>2mm}$ and WSA$_{<0.053mm}$ by 73.1% and 39.9%, and the $C_{NoF}$–$S_{CF}$ treatment decreased SOC storage of WSA$_{>2mm}$ by 39.2% compared with $C_{CF}$–$S_{CF}$ ($p < 0.05$).

**Table 5.** SOC storage (t hm$^{-2}$) within soil aggregates.

| Soil Depths | Aggregates Sizes | Treatments | | | | |
|---|---|---|---|---|---|---|
| | | $C_{NoF}$–$S_{NoF}$ | $C_{CF}$–$S_{NoF}$ | $C_{CR}$–$S_{NoR}$ | $C_{CF}$–$S_{CF}$ | $C_{CF}$–$S_{FYM}$ |
| 0–10 cm | >2 mm | 3.93 ± 0.10 b | 1.38 ± 0.04 d | 4.71 ± 0.12 a | 2.27 ± 0.06 c | 1.20 ± 0.03 d |
| | 0.25–2 mm | 11.4 ± 0.30 a | 10.9 ± 0.32 a | 11.6 ± 0.29 a | 11.1 ± 0.32 a | 10.9 ± 0.28 a |
| | 0.053–0.25 mm | 1.87 ± 0.06 c | 2.78 ± 0.09 b | 3.04 ± 0.08 b | 3.71 ± 0.12 a | 3.49 ± 0.09 a |
| | <0.053 mm | 10.1 ± 0.28 a | 7.51 ± 0.24 c | 6.95 ± 0.20 c | 7.22 ± 0.23 c | 8.51 ± 0.25 b |
| 10–20 cm | >2 mm | 1.39 ± 0.04 c | 1.78 ± 0.06 c | 1.55 ± 0.05 c | 8.33 ± 0.25 a | 2.44 ± 0.07 b |
| | 0.25–2 mm | 9.80 ± 0.31 a | 8.70 ± 0.30 b | 10.7 ± 0.33 a | 10.6 ± 0.32 a | 10.3 ± 0.30 a |
| | 0.053–0.25 mm | 1.49 ± 0.05 d | 2.73 ± 0.09 b | 3.48 ± 0.12 a | 2.49 ± 0.08 b | 1.99 ± 0.06 c |
| | <0.053 mm | 10.2 ± 0.36 a | 10.3 ± 0.35 a | 6.70 ± 0.23 c | 7.02 ± 0.23 c | 8.10 ± 0.27 b |
| 20–30 cm | >2 mm | 1.00 ± 0.03 d | 1.78 ± 0.07 b | 1.44 ± 0.05 c | 2.31 ± 0.07 a | 1.02 ± 0.03 d |
| | 0.25–2 mm | 11.8 ± 0.34 a | 10.3 ± 0.43 b | 7.31 ± 0.26 c | 12.9 ± 0.44 a | 9.62 ± 0.37 b |
| | 0.053–0.25 mm | 4.79 ± 0.14 a | 2.77 ± 0.12 c | 4.15 ± 0.16 b | 4.61 ± 0.17 ab | 3.17 ± 0.14 c |
| | <0.053 mm | 10.6 ± 0.36 a | 4.36 ± 0.25 e | 9.21 ± 0.33 b | 5.84 ± 0.26 d | 6.72 ± 0.31 c |

Note: Different lowercase letters following each value represent significant differences for each aggregate size among different treatments at the same soil layer (Tukey's HSD test, $p < 0.05$).

Additionally, the $C_{CR}$–$S_{NoR}$ and $C_{CF}$–$S_{FYM}$ treatments raised SOC content of macroaggregates (>0.25 mm) and microaggregates (<0.25 mm) by 8.89–8.41%, and 14.6–16.2% at 0–10 cm soil depth compared with $C_{CF}$-$S_{CF}$ (Figure 3). Besides, $C_{CR}$-$S_{NoR}$ increased SOC storage of macroaggregates by 22.0% relative to $C_{CF}$-$S_{CF}$.

*3.5. Correlations among Each Measured Soil Parameter*

The linear regression models among measured parameters by stepwise regression analysis are listed in Table 6. The MWD was positively correlated with WSA$_{>2mm}$ proportion (Equation (9); $R^2 = 0.855$, $p = 0$) and SOC storage in WSA$_{>2mm}$ (Equation (10); $R^2 = 0.848$, $p = 0$). Similarly, a significant and positive correlation was found between GMD and WSA$_{>2mm}$ proportion (Equation (11); $R^2 = 0.814$, $p = 0$). Meanwhile, the GMD value was positively correlated with SOC storage in WSA$_{>2mm}$ but was negatively correlated to SOC storage of WSA$_{<0.053mm}$ (Equation (12); $R^2 = 0.848$, $p = 0.027$). Additionally, significant and negative correlations were obtained between the D value and WSA$_{>2mm}$ proportion (Equation (13); $R^2 = 0.744$, $p = 0$), and between the D value and SOC storage in WSA$_{>2mm}$ proportion (Equation (14); $R^2 = 0.790$, $p = 0.002$). However, the D value was positively correlated with the WSA$_{<0.053mm}$ proportion (Equation (13)) and SOC storage in WSA$_{<0.053mm}$ proportion (Equation (14)). Based on the above data and results, we developed a diagram of the most important indicators under organic amendments, as presented in Figure 3.

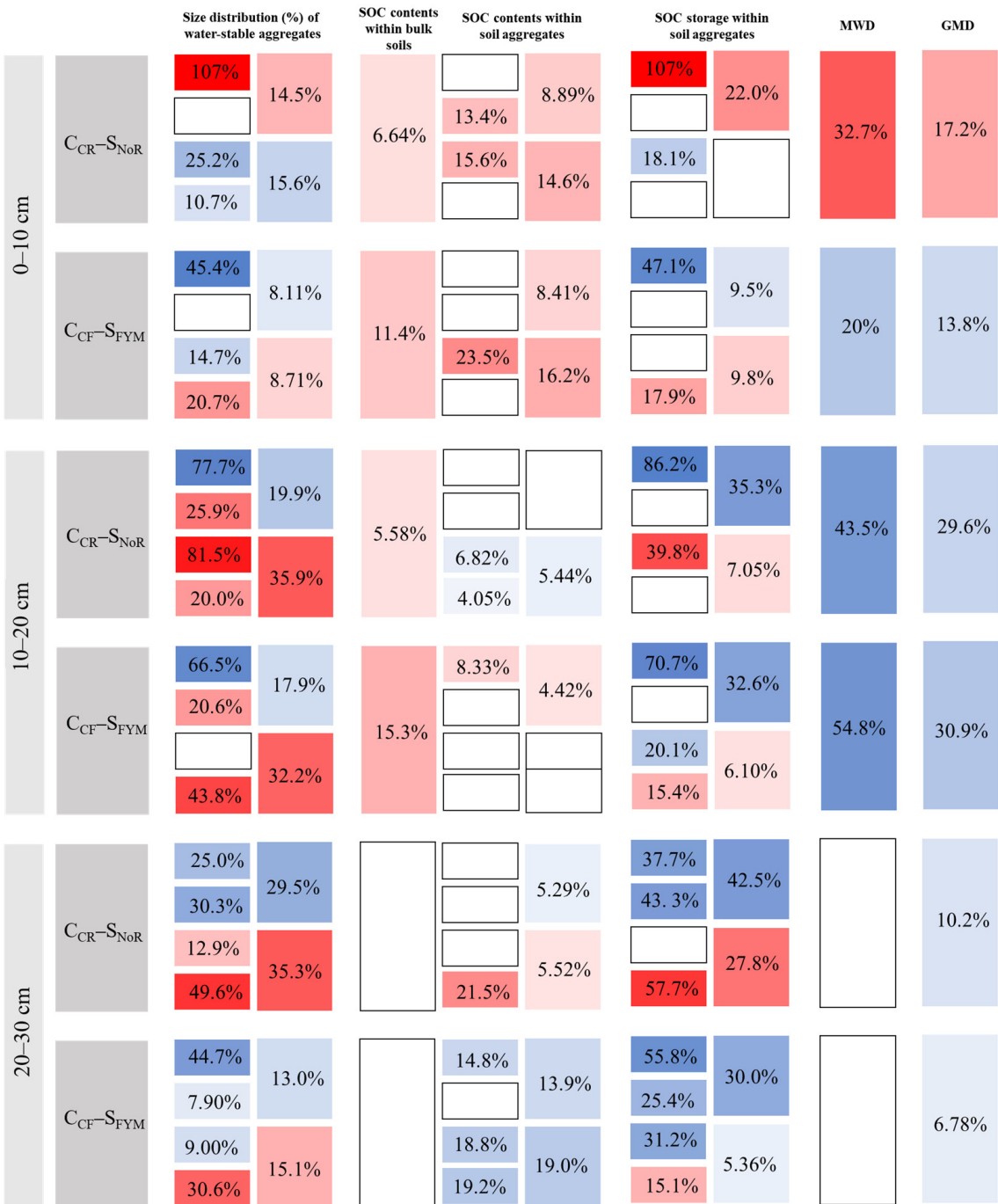

**Figure 3.** Summarized diagram of the relationship between most important indicators and soil aggregate stability under organic amendments. Note: Changes in indicators was $C_{CR}$–$S_{NoR}$ and $C_{CF}$–$S_{FYM}$ treatments compared with $C_{CF}$–$S_{CF}$ treatment. Red and blue blocks represent increment and decrement, respectively, and the shade color represents the change amplitude. The four blocks in a vertical row from top to bottom represent >2 mm, 2–0.25 mm, 0.25–0.053 mm, and <0.053 mm sized aggregates. The two blocks in a vertical row from top to bottom represent macroaggregates (>0.25 mm) and microaggregates (<0.25 mm).

**Table 6.** Correlations among each measured soil parameter.

| Indicators in Y | Indicators in X | Regression Model | $R^2$ | F | P |
|---|---|---|---|---|---|
| Aggregate stability (MWD) in water–stable aggregates | The proportion of aggregate size fractions (ASF) in water–stable aggregates particle sizes | $Y_{MWD} = 0.055\ ASF_{WSA>2mm} + 0.596$ (9) | 0.855 | 83.649 | 0 |
| Aggregate stability (MWD) in water–stable aggregates | Soil organic carbon storage (SOCS) in water–stable aggregates particle sizes | $Y_{MWD} = 0.183\ SOCS_{WSA>2mm} + 0.661$ (10) | 0.848 | 78.978 | 0 |
| Aggregate stability (GMD) in water–stable aggregates | The proportion of aggregate size fractions (ASF) in water–stable aggregates particle sizes | $Y_{GMD} = 0.012\ ASF_{WSA>2mm} + 0.476$ (11) | 0.814 | 62.118 | 0 |
| Aggregate stability (GMD) in water–stable aggregates | Soil organic carbon storage (SOCS) in water–stable aggregates particle sizes | $Y_{GMD} = 0.036\ SOCS_{WSA>2mm} - 0.011\ SOCS_{WSA<0.053mm} + 0.584$ (12) | 0.848 | 6.359 | 0.027 |
| Aggregate stability (D) in water–stable aggregates | The proportion of aggregate size fractions (ASF) in water–stable aggregates particle sizes | $Y_D = -0.003\ ASF_{WSA>2mm} + 0.003\ ASF_{WSA<0.053mm} + 2.751$ (13) | 0.744 | 21.337 | 0 |
| Aggregate stability (D) in water–stable aggregates | Soil organic carbon storage (SOCS) in water–stable aggregates particle sizes | $Y_D = -0.014\ SOCS_{WSA>2mm} + 0.009\ SOCS_{WSA<0.053\ mm} + 2.782$ (14) | 0.790 | 14.521 | 0.002 |

Note: $R^2$ is the coefficient of determination, and it indicates the explanation rate of independent variable for dependent variable; F is a test of variance for the regression model as a whole; Significance tests for coefficients of individual variables generally up to *p* value, if the *p* value < 0.05, the effect of the independent variable is significant, *n* = 15.

## 4. Discussion

Soil aggregate stability is a vital index for evaluating soil quality [38]. The aggregate sized with >0.25 mm plays a crucial part in sustaining the stability of soil structure and is considered the best structure of the soil [23]. For the five treatments in the present study, we found that 0.25–2 mm sized aggregates exhibited the biggest proportions, which indicates that Mollisols have more smaller macro–aggregates as its ideal structure.

MWD, GMD, and D values are all parameters of soil aggregate stability. The greater MWD and GMD values, and the smaller D value mean the stronger stability of soil structure [39]. In our experiment, the significantly positive correlation between MWD/GMD and the $WSA_{>2mm}$ proportion, and between D and the $WSA_{<0.053mm}$ proportion, as well as the significantly negative relationship between D and the $WSA_{>2mm}$ proportion all demonstrated that macroaggregates (>2 mm) and silt + clay (<0.053 mm) fractions contributed the most to the stability of soil aggregates. This association might be due to macroaggregates developed by surrounding new inputs of organic carbon as an additional source for microbial activity, thus promoting the formation of aggregate binders, resulting in a larger value of MWD and GMD [40]. Additionally, the relationship of strong linearity between MWD/GMD and SOC storage in $WSA_{>2mm}$ suggested that SOC storage in the $WSA_{>2mm}$ exhibited a vital part in soil aggregate stability. However, Das et al. believed that MWD as one of the soil aggregate stability indicators was related to SOC content [41]. The discrepancy might be because of the different soil type, substrate type, and the natural conditions of experimental site and so on.

The current research indicated that crop residue return increased soil aggregate stability at 0–10 cm soil depth, but manure addition decreased it at 0–20 cm soil depth. The increased soil aggregate stability under crop residues at 0–10 cm compared to continuous

chemical fertilizer application could be due to the increment proportion of $WSA_{>2mm}$ and the decrement proportion of $WSA_{<0.053mm}$ at 0–10 cm soil depth in the current research. In comparison to continuous chemical fertilized crops, improved crop growth [42] by incorporation of crop residues and chemical fertilizer enhanced the return of organic matter, thus raising SOC content, which is the main binding agent of aggregates, leading microaggregates combined into macroaggregates [43]. In addition to improving crop growth causing a range of effects on macroaggregates, the carbohydrates produced during the humification process after crop residues strengthen soil microbial activity and promote plant root vigor and humus formation, which has a profound positive impact on the formation of macroaggregates, and thus larger values of MWD and GMD (Figure 2) [26,44].

The decreased soil aggregate stability under manure addition at 0–20 cm compared to continuous chemical fertilizer application could be due to the increment proportion of $WSA_{<0.053mm}$ and the decrement proportion of $WSA_{>2mm}$ in the present research. Silt + clay fraction ($WSA_{<0.053mm}$) proportion was crucial determinants of water stable aggregation [45,46]. Accordingly, higher silt + clay concentration aggregates are less susceptible to destructive forces than lower silt + clay concentration ones [47,48]. In addition, the increased moisture content in the current research promoted microbial activity, which enhanced the breakdown of macroaggregates [49,50]. The negative correlation between soil moisture content and aggregate stability was because soil minerals swell unevenly when wet, which induces aggregate cracking [51,52]. Therefore, increasing drainage should be considered to solve this problem under corn–soybean rotation in Mollisols.

SOC content is an important parameter that affects soil physical properties. In our study, the 6–yr application of manure and crop residues both raised the SOC content compared to chemical fertilizer application at 0–20 cm. The order of SOC content was manure addition > crop residue return. It is obvious that crop residue return and manure application are organic carbon inputs themselves. In addition, crop residue return can improve C content by modulating C–related microbial abundance. Manure usually has a high SOC sequestration efficiency [53]. The meta–analysis of 95 studies showed that crop residue return significantly increased the SOC content of Chinese farmland by an average of 13.97% [54]. They further proposed that the increased SOC by crop residue return would be less pronounced in the region with a mean annual temperature < 10 °C or initial SOC content greater than 10 g kg$^{-1}$. The mean annual temperature in our experiment site is 2.4 °C, and the SOC content only increased by 6.23%. Low temperature reduces the activity of straw–degrading microorganisms [55,56]. The application of manure elevated SOC content at different rates, and the limitations of other factors were not obvious [21,57,58]. Thus, the 6–yr application of manure is the best way to enhance SOC content in Chinese Mollisols, while crop residues returning follows behind.

## 5. Conclusions

We investigated 6–yr different fertilization practices on aggregate size distribution, SOC content, and storage within soil aggregates and bulk soil of Mollisols under corn–soybean rotation. Compared with consecutive application of chemical fertilizer alone, annual alternate application of manure and chemical fertilizer increased bulk SOC by 13.4%, but reduced MWD and GMD values by 37.4% and 22.4% at 0–20 cm. Simultaneously, the combined application of consecutive returning crop residues and chemical fertilizer in alternate years increased bulk SOC by 6.23% at 0–20 cm, and increased MWD and GMD values by 32.7% and 17.2% at 0–10 cm. No significant differences were presented for MWD values among the five treatments at 20–30 cm soil depth. Altogether, we infer that the combined application of consecutive returning crop residues and chemical fertilizer in alternate years is the most favorable approach for SOC accumulation and aggregate stability at 0–10 cm soil depth under corn–soybean rotation in Mollisols.

**Author Contributions:** Conceptualization, Y.X., M.Z. and X.L.; methodology, Y.X., M.Z. and Y.L.; software, Y.X., M.Z. and X.Z. (Xiannan Zeng); validation, M.Z.; formal analysis, Y.X. and M.Z.; investigation, Y.L. and X.Z. (Xingyi Zhang); resources, X.Z. (Xingyi Zhang), G.W., J.J. and X.L.; data curation, Y.X. and M.Z.; writing—original draft preparation, Y.X.; writing—review and editing, M.Z., G.D. and X.L.; visualization, Y.X.; supervision, G.W. and X.L.; project administration, Y.L. and X.L.; funding acquisition, X.L. All authors have read and agreed to the published version of the manuscript.

**Funding:** This research was funded by the National Key R&D Program of China (2021YFD1500700), Professional Association of the Alliance of International Science Organizations (ANSO–PA–2020–12), Strategic Priority Research Program of the Chinese Academy of Sciences (XDA28070302), and the fund for Heilongjiang Scientific Research Institutions (CZKYF2021–2–C027).

**Institutional Review Board Statement:** Not applicable.

**Informed Consent Statement:** Not applicable.

**Data Availability Statement:** The data that support the findings of this study are available from the corresponding author upon reasonable request.

**Conflicts of Interest:** The authors declare no conflict of interest.

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
