# Peer review of "Crop Residue Return Rather Than Organic Manure Increases Soil Aggregate Stability under Corn–Soybean Rotation in Surface Mollisols"

_agriculture, doi:10.3390/agriculture12020265_

Round 1

Reviewer 1 Report

I read with a great interest the manuscript "Straw return rather than organic manure increases soil aggregates stability under corn-soybean rotation in surface Mollisols".  The stability of agregates is important for soil structure and at the end soil fertility. According to my meaning the manuscript is written clearly and conclusions are supported by obtained results. In the text you will find some my reccomendations to improve the text. Some clarifications should be given in  mainly in methodology. I am not sure if you really measured SOC and not total C (Ct). Also the doses of nutrients should be given in kg of nutrient per ha (N, P, K) and not the total dose of fertilizer.

Reviewer 2 Report

General comment: corn and soybean are macro-thermal crops with vegetation zero around 10 °C and therefore it is impossible to grow them with an average annual temperature of 1.5 °C. Please, discuss this point.

In particular, the following necessary improvements are:

The complete manuscript needs to be revised according to Functional the instructions for authors.

Rewrite the abstract: no use of abbreviations.

It is recommended to accurately review the English form, the language is not fluent and the sentences are disconnected.

Page 1, line 40: “Aggregates are hierarchical in soil”:  explains the content of the sentence.

Page 2, 91-93: The site is under semi-humid continental monsoon climate condition with effective accumulated temperature of  2450 ℃, and the average annual rainfall and temperature are 553.9 and 1.5 ℃, respectively. Comment: The accumulated annual temperature of  2450 ℃ requires an average annual temperature of 20 °C and certainly not 1.5 °C. Please, discuss this point.

Table 1. Basic properties of soil fertility in experimental area. Comment: report the methods of analysis; soil organic matter, delete soil; available potassium or exchangeable potassium?

Page 3, line 138, 139: Ring knife was applied in the same sampling points and depths for calculating bulk density, SOC and total nitrogen (TN) contents. Comment : cutting ring (diameter ?;  volume ?); describe better the method used to measure  soil bulk density. On what fraction of soil was SOC and TN determined? On fine soil (<2 mm)?.

Page 4, line 141-142: About 10 g of soil sample was taken and weighed, then was dried at 105 ± 2℃ to  constant weight.   Comment: The classical procedure of gravimetric method used for the measurement of soil moisture involves collecting a  core  samples of   40-50 g of soil. 10 g of soil is not significant.

Reviewer 3 Report

My comments on the manuscript Straw Return Rather Than Organic Manure Increases Soil Aggregates Stability under Corn-Soybean Rotation In Surface Mollisols, which has been submitted to Agriculture journal, are presented below.

The manuscript is very interesting. The Authors comprehensively presented the problem of enhancing soil quality and mechanisms involved between SOC and aggregates stability.

In my opinion, the work was written very carefully. The Authors of the study report the sowing density for soybean 250,000 ha−1 and corn 48,000 ha−1. Please explain why this seeding density was used. Please complete the information in the methodology and specify the row spacing for corn and soybean.

Reviewer 4 Report

Interesting manuscript, giving valuable information about impact of different fertilization regimes and crop rotation on soil structural agregates.

Language editing is required.

Style errors are present in text.

Most of the suggestions are present in text.

Round 2

Reviewer 2 Report

Dear Ms. Patricia Li

Section Managing Editor

Agriculture – MDPI

I send the revision response concerning the Manuscript ID: Agriculture-1576355-peer-review-v2, entitled " Crop residues straw return rather than organic manure increases soil aggregates stability under corn-soybean rotation in surface Mollisols”.

General comments

I examined the manuscript all the suggested changes have been carried out correctly, including English form, figures and tables and their captions.

The paper can be accepted for publication.

Sincerely

Ph.D Giovanni Lacolla

This manuscript is a resubmission of an earlier submission. The following is a list of the peer review reports and author responses from that submission.